# Single-Cell Transcriptomic Profiling Unveils Dynamic Immune Cell Responses during *Haemonchus contortus* Infection

**DOI:** 10.3390/cells13100842

**Published:** 2024-05-15

**Authors:** Wenxuan Wang, Zhe Jin, Mei Kong, Zhuofan Yan, Liangliang Fu, Xiaoyong Du

**Affiliations:** 1College of Informatics, Huazhong Agricultural University, Wuhan 430070, China; 2020317110026@webmail.hzau.edu.cn (W.W.); dare@webmail.hzau.edu.cn (Z.J.); mkong@webmail.hzau.edu.cn (M.K.); zfyan@webmail.hzau.edu.cn (Z.Y.); 2Key Laboratory of Smart Animal Farming Technology, Ministry of Agriculture and Rural Affairs, Huazhong Agricultural University, Wuhan 430070, China; 3Key Laboratory of Agricultural Animal Genetics, Breeding and Reproduction, Ministry of Education, College of Animal Science and Technology, Huazhong Agricultural University, Wuhan 430070, China; fuliangliang2017@hzau.edu.cn

**Keywords:** single-cell transcriptomics, goats, *Haemonchus contortus*, parasitic resistance, peripheral blood mononuclear cells

## Abstract

Background: *Haemonchus contortus* is a parasite widely distributed in tropical, subtropical, and warm temperate regions, causing significant economic losses in the livestock industry worldwide. However, little is known about the genetics of *H. contortus* resistance in livestock. In this study, we monitor the dynamic immune cell responses in diverse peripheral blood mononuclear cells (PBMCs) during *H. contortus* infection in goats through single-cell RNA sequencing (scRNA-Seq) analysis. Methods and Results: A total of four Boer goats, two goats with oral infection with the L3 larvae of *H. contortus* and two healthy goats as controls, were used in the animal test. The infection model in goats was established and validated by the fecal egg count (FEC) test and qPCR analysis of the gene expression of IL-5 and IL-6. Using scRNA-Seq, we identified seven cell types, including T cells, monocytes, natural killer cells, B cells, and dendritic cells with distinct gene expression signatures. After identifying cell subpopulations of differentially expressed genes (DEGs) in the case and control groups, we observed the upregulation of multiple inflammation-associated genes, including NFKBIA and NFKBID. Kyoto Encyclopedia of the Genome (KEGG) enrichment analysis revealed significant enrichment of NOD-like receptor pathways and Th1/Th2 cell differentiation signaling pathways in CD4 T cells DEGs. Furthermore, the analysis of ligand–receptor interaction networks showed a more active state of cellular communication in the PBMCs from the case group, and the inflammatory response associated MIF–(CD74 + CXCR4) ligand receptor complex was significantly more activated in the case group, suggesting a potential inflammatory response. Conclusions: Our study preliminarily revealed transcriptomic profiling characterizing the cell type specific mechanisms in host PBMCs at the single-cell level during *H. contortus* infection.

## 1. Introduction

*Haemonchus contortus* is widely distributed in tropical, subtropical, and warm temperate regions, where it primarily parasitizes the abomasum of various ruminants and feeds on the host’s blood, with the parasitism significantly endangering the health of these animals [1]. The main strategy for the prevention and treatment of *H. contortus* infections has traditionally been the use of chemical drugs. However, the increasing drug resistance of the parasite highlights the urgent need within the livestock industry for alternative strategies to combat this infection [2,3,4,5]. In addition, the development of more effective means of control measures requires a better understanding of the immune response in goats infected with *H. contortus*.

The innate immune system represents the first line of defense for the host, acting as a crucial initial barrier against pathogens and other foreign substances. It also plays a crucial role in the activation and regulation of the adaptive immune response [6]. Composed of various cells and molecules, including macrophages, dendritic cells (DCs), natural killer (NK) cells, neutrophils, the complement system, and related receptors and inflammatory mediators, the innate immune system responds rapidly and effectively to a wide range of pathogens and insults [7]. Furthermore, innate immune pattern recognition receptors (PRRs) of the innate immune system play an important role in parasitic infections. Research has highlighted the role of specific immune receptors, such as Toll-like receptors (TLRs) and NOD-like receptors (NLRs), in recognizing pathogen-associated molecular patterns (PAMPs) of helminths, thereby orchestrating the host immune response [8,9]. These receptors promote a series of immune responses, including the infiltration of inflammatory cells, the activation of NK cells, and the production of antibodies, thereby enhancing the host’s defense capabilities against worm invasion and parasitism [10]. Current research on the interaction between *H. contortus* and host pattern recognition receptors is still in its infancy, with only a few studies involving CLRs, NLRs, and TLRs [11,12,13].

Additionally, parasites have developed complex mechanisms to evade the host’s immune response, such as altering their surface molecules and releasing inhibitory factors [14]. Using the excretory-secretory proteins of *H. contortus* is a key strategy for immune evasion, with extensive research already conducted in this area. A previous study identified 114 *H. contortus* excretory-secretory proteins (HcESPs) that interact with host T cells, finding that HcESPs can suppress T cell viability, induce apoptosis, inhibit T cell proliferation, and cause cell cycle arrest [15]. Other studies have found that proteases, ES-15 kDa protein, ES-24 kDa protein, ES-55 kDa protein, and arginine kinase secreted by *H. contortus* can interact with the host’s peripheral blood mononuclear cells (PBMCs), either stimulating or inhibiting the host immune cell’s immune response [16]. PBMCs derived from the bone marrow and lymphoid hematopoietic systems consist mainly of circulating multifunctional immune cell types, such as lymphocytes (including T cells, B cells, and NK cells), monocytes, and DCs [17]. As critical mediators and effectors of the body’s defense mechanisms, PBMCs play a vital role in mediating innate and adaptive immune responses, maintaining immune homeostasis, and reflecting the real-time cellular and humoral immune status of the body. Previous studies have indicated that *H. contortus* can induce an anti-inflammatory immune response in the host’s PBMCs [12]. However, the roles of and synergistic interactions among PBMC subpopulations in the immune response are still unclear.

In this study, through single-cell RNA sequencing (scRNA-Seq) analysis of PBMCs from goats infected with *H. contortus* and healthy goats, we identified and annotated subpopulations of goat PBMCs. Using differential gene expression analysis, Kyoto Encyclopedia of the Genome (KEGG) enrichment analysis, cell communication analysis, and additional methodologies, we explored the immune mechanisms of different PBMC subpopulations during *H. contortus* infection and their synergistic interactions. The results provide valuable information for further understanding the immune response mechanisms between goats and *H. contortus*.

## 2. Materials and Methods

### 2.1. Sample Collection

Healthy goats from the same herd at 4 months of age, after administration of anthelmintics and confirmation of the absence of parasitic infection by fecal egg examination, were each kept separately for 28 days, and then the absence of infection was reconfirmed by fecal egg examination. Two goats were each orally inoculated with 12,000 *H. contortus* L3 larvae. As a control, two goats received an equivalent volume of water orally. PBMCs were isolated at room temperature using lymphocyte separation (Tianjin Haoyang Biological, Tianjin, China), and cell viability was assessed via 0.4% trypan blue staining (Shenggong Biological, Shanghai, China). The care and management of the experimental animals were conducted in compliance with the Hubei Province regulations on experimental animals, with the approval of the Experimental Animal Administration and Ethics Committee of Huazhong Agricultural University.

### 2.2. Fecal Egg Count Detection

Eighteen days following infection with *H. contortus*, fecal samples from the goats were collected for egg quantification using the saturated salt water flotation method. From the initial detection of eggs, the number of eggs per gram (EPG) of feces were determined twice daily using the McMaster egg counting technique.

### 2.3. Quantitative Real-Time PCR

Total RNA was extracted from cellular samples using the TRIzol reagent (Invitrogen, Carlsbad, CA, USA) according to the manufacturer’s instructions. Complementary DNA (cDNA) was synthesized using the HiScript III All-in-one RT SuperMix Perfect for qPCR kit (Vazyme, Nanjing, China). qPCR assays were performed in triplicate on a Bio-Rad qPCR instrument using the Taq Pro Universal SYBR qPCR Master Mix (Vazyme, Nanjing, China). Primer sequences are provided in the Appendix A.

### 2.4. Single-Cell RNA-Seq Library Preparation and Sequencing

Berry Genomics Co., Ltd., Beijing, China, assisted in single-cell transcriptome library construction and sequencing. The process involved using the Chromium Single Cell 3’ V3 Gel Beads kit from 10×Genomics and mixing a diluted PBMC suspension with Master Mix to form Gel Bead-In-EMulsions for RT-PCR amplification. After enzymatic digestion of the amplified products for fragmentation, sequencing adaptors were added, followed by PCR amplification to introduce sample indexes, constructing cDNA libraries with P5 and P7 adapters. Libraries meeting quality standards were sequenced on the NovaSeq 6000 system (Illumina) using the PE150 mode for paired-end sequencing.

### 2.5. Single-Cell RNA-Seq Data Analysis

Sequencing data were aligned to the goat reference genome (ARS1) using CellRanger (v3.1.0), with gene expression quantification based on reference genome GTF file annotations. Subsequent analyses were performed using the Seurat (v3.2) R package.

### 2.6. Quality Control, Dimension Reduction, and Cell Clustering

Statistical analyses were conducted for each sample to obtain UMI counts, gene quantity, and the mitochondrial gene percentage. Low-quality cells and genes were excluded: genes expressed in fewer than 3 cells, cells with total gene counts outside the range of 500 to 4000, cells with RNA counts outside the range of 1000 to 20,000, and cells with a mitochondrial gene proportion exceeding 20% were removed. Doublets were identified and eliminated using the DoubletFinder (v2.0.3) R package [18]. Data normalization used the log normalization method to negate library size effects, and gene expression was standardized using ScaleData() to minimize disparities in gene expression magnitudes. The “vst” method via the FindVariableFeatures() function identified highly variable genes (HVGs), identifying genes with average expression levels between 1 and 10 and a dispersion greater than 1 as HVGs. The top 2000 genes were selected as HVGs for PCA dimensionality reduction on the normalized expression matrix. The ElbowPlot() function from Seurat determined the significant number of principal components (PCs). Using the FindNeighbors() function from Seurat, shared nearest-neighbor (SNN) graphs were built based on Euclidean distances in PCA space, and unsupervised clustering was executed on this SNN with the “Louvain” algorithm through the FindClusters() function. Batch effects were addressed by data integration using Harmony (v0.1.0) [19]. For enhanced visualization, the UMAP method applied via the RunUMAP() function from Seurat reduced dimensionality, using the same count of PCs as that used for clustering.

### 2.7. Cell Subpopulation Annotation

Marker genes for subpopulations were identified using the Wilcoxon rank sum test in the FindAllMarkers() function from the Seurat package, with a threshold set to a log2 fold change ≥ 0.4 and expression in at least 20% of the subpopulation. Bonferroni correction was applied to control the false discovery rate (FDR), with genes having an adjusted *p*-value < 0.05 considered as markers for subpopulations. Cell marker genes were referenced from the PanglaoDB database [20] (https://panglaodb.se/) and the CellMarker database [21] (http://biocc.hrbmu.edu.cn/CellMarker (accessed on 9 October 2023)), with the reference species being humans. Auxiliary annotation of PBMC samples was performed using the SingleR package (v 2.2.0), with the BlueprintEncodeData dataset as a reference [22]. Subpopulations were manually annotated by combining subpopulation marker genes with SingleR annotation results.

### 2.8. Cell Cycle Analysis

We calculated the potential cell cycle phase of each cell by analyzing the expression of classic marker genes for the G2M and S phases. These marker gene sets and their expression levels are inversely related, with cells not expressing these marker genes classified as being in the G1 phase. Using the CellCycleScoring() function from the Seurat package, scores for each cell in the S and G2M phases were calculated, allowing for the classification of cells into distinct cell cycle stages.

### 2.9. Gene Differential Expression Analysis

The Wilcoxon rank sum test in the FindMarkers() function was used for gene differential expression analysis between control and case groups for each subpopulation. A threshold of a log2 fold change greater than or equal to 0.25 and expression in at least 20% of the subpopulation was set, with Bonferroni correction applied to control the FDR. An adjusted *p*-value less than 0.05 was considered to indicate genes with significant differential expression. GO analysis and KEGG enrichment analysis were performed using clusterProfiler, with a selection threshold of *p*-value < 0.05.

### 2.10. SCENIC Analysis

The R package Single-Cell Regulatory Network Inference and Clustering (SCENIC) (v 1.1.2) [23] was used to predict the main transcription factors and gene regulatory networks in PBMCs’ scRNA-Seq data under default parameters. SCENIC analyzes co-expression patterns between transcription factors and target genes using machine learning algorithms, combined with enrichment analysis of transcription-factor-binding sites, to infer potential regulatory relationships. In summary, SCENIC identifies potential target genes of transcription factors and constructs co-expression modules using the GENIE3 method. Regulons with significant DNA-motif enrichment of direct targets are selected using the RcisTarget tool, forming regulatory elements. The activity of regulons in each cell is assessed using AUCell by calculating the area under the curve (AUC) of gene expression values to determine the activation status of regulons. In this study, we analyzed T cells and monocytes from control and case groups to identify key transcription factors and regulatory mechanisms.

### 2.11. Cell Communication Analysis

Cellular communication analysis was performed using the R package CellChat [24]. A normalized gene expression matrix and cell type annotation information were extracted from the Seurat object. Using the CellChat tool and single-cell transcriptomics expression matrix data, the enrichment level of receptor–ligand pairs between two cell types was predicted based on the expression of receptors in one cell type and ligands in another. In summary, a CellChat object is constructed using the normalized single-cell expression matrix. Overexpressed ligands or receptors in the cell group are detected, and their interactions are further identified. The expression ratio of receptor/ligand genes for each cell type is calculated, and genes expressed in more than 10% of cells are included in further analysis, along with the calculation of the mean gene expression. By leveraging known interactions between signaling ligands, receptors, and co-factors, combined with gene expression data and cell–cell communication patterns, the probability of cell–cell communication is simulated, assigning a probability value to each interaction. Permutation tests are conducted to infer biologically meaningful cell–cell communications for visualization analysis.

## 3. Results

### 3.1. Parasite Load and Immune Response in H. contortus-Infected Goats

We analyzed two goats infected with *H. contortus* as cases and two healthy goats as controls (Figure 1A). After the goats were infected with *H. contortus*, eggs were first detected in the goats’ feces on day 20, with an average of 419 EPG. Subsequently, the number of eggs continued to increase and peaked on day 30, with an average of 3034 EPG of feces. After the peak, there was a decline in the number of eggs, with counts on day 32 averaging 2588 EPG and on day 34 averaging 2650 EPG (Figure 1B). This indicates successful oral infection of the goats with the L3 larvae of *H. contortus* and that the parasites had established and reproduced within the goats. Subsequently, we isolated PBMCs for qPCR and scRNA-Seq analysis, with all samples showing over 85% cell viability (Appendix A).

Dynamic changes in immune gene expression were observed in goat PBMCs following *H. contortus* infection. Immune gene expression showed no significant changes on days 3 and 7 postinfection. On day 10, IL-5 expression increased significantly (1.84-fold, *p*-value < 0.05) and peaked on day 14 (2.36-fold, *p*-value < 0.01), along with a significant increase in IL-6 expression (1.54-fold, *p*-value < 0.05). IL-6 levels peaked on day 21 (1.8-fold, *p*-value < 0.05), and by day 30, all immune gene expressions had stabilized, indicating a diminishing immune response to *H. contortus* infection (Figure 1C). Cytokines are pivotal mediators of immune responses, and the host can stimulate the production of Th1 and Th2 cytokines, such as IL-4, IL-5, IL-6, IL-10, and IL-13, to combat infections [25,26,27]. Among these, IL-5 and IL-6 are the most critical cytokines in nematode infections. In this study, the significantly higher expression of IL-5 and IL-6 in the case group’s goat PBMCs on day 14 compared to that in the control group indicates that the infection effectively stimulated the host immune response, resulting in increased expression levels of cytokines, such as IL-5 and IL-6. This also indirectly suggests the successful establishment of a goat model infected with *H. contortus*.

### 3.2. Annotation of Subpopulations in Goat PBMCs

To elucidate the immune response mechanisms of goat PBMCs during *H. contortus* infection, we conducted scRNA-Seq analysis in goat PBMCs and annotated the cell subpopulations. This comprehensive analysis yielded data from a total of 31,756 cells, averaging 62,254 reads per cell. The alignment rate to the sample genome exceeded 90% for all samples, and we detected a total of 16,771 genes, with an average of 1302 genes per cell. After excluding low quality cells, we obtained 7107 cells from the C1 sample, 8041 cells from the C2 sample, 7418 cells from the T1 sample, and 7865 cells from the T2 sample (Appendix A). We selected the top 2000 highly variable genes for principal component analysis (PCA) to reduce dimensionality (Appendix A), and an elbow plot based on the principal components’ variance percentage ranking was generated. Combining the elbow plot with analysis results, 25 principal components were used for further analysis (Appendix A). Using the Harmony algorithm to correct batch effects, we achieved clear clustering, delineating 21 distinct cellular clusters (Figure 2A).

Based on the known marker genes, we made preliminary annotations for these subgroups. Additionally, the SingleR package was used to automatically annotate the cell subgroups (Figure 2B). Combining automatic annotation results with typical marker genes for immune cell populations, 20 cell subpopulations were manually annotated into seven cell types: T cells (CD3E), B cells (MS4A1, CD79), NK cells (NKG7, GNLY, GZM), monocytes (CD14, CFP), DCs (FLT, TREM2), Plasmablasts (MZB1, JCHAIN), and platelets (GNG11, NRGN). We excluded cells expressing multiple marker genes and generated annotated UMAP plots (Figure 2C). Additionally, the expression of marker genes across various cell subpopulations confirmed the accuracy of the annotations (Figure 2D).

Considering the critical role of T cells in combating parasitic infections, we further dissected this population, identifying 21 unique T cell subpopulations. Similar annotation procedures were applied, resulting in the annotation of 20 cell subpopulations into 10 T cell subtypes: CD4CD8 double-negative T cells, naive CD4+ T cells, proliferating CD4+ T cells, CD4 central memory T cells (TCM), CD4 effector memory T cells (TEM), naive CD8+ T cells, proliferating CD8+ T cells, and CD8 TCM, CD8 TEM, and regulatory T cells (Treg). Cells with multi-marker gene expression were excluded, followed by the generation of an annotated UMAP plot (Figure 2E). The expression of T cell-subpopulation-specific marker genes was visualized using bubble plots, revealing uniformly high expression across all subpopulations (Figure 2F). This detailed classification and subsequent analysis elucidated the intricate immune landscape within goat PBMCs during *H. contortus* infection.

### 3.3. Infection with H. contortus Alters the Gene Expression and Proportions of PBMCs

To investigate cell-type-specific transcriptional alterations, we identified differentially expressed genes (DEGs) across various immune cells: CD4 T cells, CD8 T cells, B cells, monocytes, DCs, and NK cells. The analysis revealed 693 DEGs in CD4 T cells (331 upregulated, 362 downregulated), 761 in CD8 T cells (316 upregulated, 445 downregulated), 584 in B cells (277 upregulated, 307 downregulated), 848 in monocytes (442 upregulated, 406 downregulated), 945 in DCs (402 upregulated, 543 downregulated), and 791 in NK cells (357 upregulated, 434 downregulated) (abs (log2 fold change) ≥ 0.25 and *p*-value < 0.05). We observed a higher number of downregulated genes in all subpopulations except for monocytes. We ranked the top 10 most upregulated and downregulated genes in each cellular subpopulation (Figure 3A–F) and noted the upregulation of inflammation-related genes, such as NFKBIA and NFKBID, suggesting that infection with *H. contortus* may induce an inflammatory response in the host. Notably, DCs had the highest number of downregulated DEGs, totaling 543, which may be associated with the immunosuppressive effects of *H. contortus* on immune cells. Further analyses, including Gene Ontology (GO) and KEGG enrichment, were performed on the DEGs of each cell subpopulation. The KEGG analysis revealed significant enrichment of various PRR signaling pathways, including NLRs and TLRs (Figure 3G), hinting at their crucial role in mediating the immune response against *H. contortus* infection. Concurrently, GO analysis revealed noteworthy enrichment in terms related to the innate immune response, cell cycle, and regulation of cell communication, indicating that *H. contortus* infection prompts a robust innate immune activation, while influencing cell cycle progression and intercellular communication within the host immune system (Figure 3H).

We also evaluated alterations in the composition of PBMC subpopulations and cell cycle dynamics between the case and control groups. Notably, the ratio of CD8 T cells to NK cells escalated in the infected animals. Specifically, CD8 T cells increased from 11.66% to 16.06% in the control group, whereas NK cells rose from 4% to 7.61%. Conversely, the percentage of monocytes decreased, dropping from 10.29% in the control group to 5.47% in the case group (Appendix A). Regarding cell cycle variations, T cells and NK cells in the case group exhibited a higher proportion in the G1 phase and a lower proportion in the G2M phase compared to the control group. However, the proportions of B cells, monocytes, and DCs were comparable in both G1 and G2M phases, suggesting that the proliferative activity of T cells and NK cells was attenuated in the case group. This finding implies that *H. contortus* may exert inhibitory effects on the proliferative capacity of immune cells in goats. Furthermore, our study revealed that among the five types of immune cells examined, B cell and T cell subpopulations exhibited the lowest proportion of cells in the G2M stage. This indicates that these cell subpopulations may primarily exist in a quiescent state, exhibiting limited proliferative potential. Conversely, the NK cell subpopulation displayed a higher proportion of cells in the G2M stage, suggesting that these subpopulations are more actively engaged in the division phase and possess a greater proliferative capacity (Appendix A).

In conclusion, our study unveils the complex transcriptional changes and pathway activations across specific immune cell subpopulations in goats infected with *H. contortus*. These findings highlight the dynamic and multifaceted nature of the host immune response, offering valuable insights into the immunological strategies mobilized against *H. contortus* infection.

### 3.4. Infection with H. contortus Enhances Intercellular Communication in PBMCs

To investigate the intercellular communication network in PBMCs of goats infected with *H. contortus*, we performed cellular communication analysis on scRNA-Seq data and constructed a network. The analysis revealed a denser network of ligand–receptor interactions, with increased interaction intensities within the PBMCs from the case group compared to those from the control group (Figure 4A–D). This observation underscores a more active cellular communication landscape in goats infected with *H. contortus*, highlighting an enhanced state of immune response activation.

In addition, we noted that the CD4 T and NK cell subpopulations exhibited the greatest differences between the control group and the case group, with the case group showing enhanced intercellular interactions between T cell and NK cell subpopulations, suggesting their pivotal role in the immune response to the parasite (Appendix A). To elucidate the interactions among various cell subpopulations within the cellular communication network of PBMCs, the top 10 most significant signaling pathways in the communication network were identified. These included tumor necrosis factor (TNF), transforming growth factor-β (TGF-β), Bcl-2-associated athanogene (BAG), Fms-like tyrosine kinase-3 (FLT3), interleukin-16 (IL-16), chemokines, CD40 ligand complement protein, resistin, macrophage migration inhibitory factor (MIF), and B-cell-activating factor (BAFF). By considering all types of cellular groups from the control and case group goats’ PBMCs as signal senders or receivers, all interactions between each cell group and other groups were obtained. As shown in Figure 4E,F, the TNF–TNFRSF1A, TNF–TNFRSF1B, IL16–CD4, FLT3L–FLT3, and CXCL8–CXCR2 ligand–receptor interactions were unique to the case group, suggesting that the immune response of the case group goats to *H. contortus* may be facilitated by signaling pathways or cytokines related to TNF, IL16, and FLT3. Additionally, in the MIF–(CD74 + CXCR4) receptor–ligand interaction, the interactions between B cells and DCs, CD4 T cells and B cells, CD4 T cells and DCs, and CD8 T cells and DCs were stronger in the case group than in the control group. We examined the changes in MIF-based cell communication in the case group and found that there was a higher number of MIF cell communication events in the case group, with a greater number of cell subpopulations involved in MIF cell communication (Figure 4G,H). This comprehensive exploration enriches our understanding of the dynamic and complex immune interactions between goats and *H. contortus.*

### 3.5. Transcriptional Factor Activation during H. contortus Infection

We analyzed single-cell transcriptional regulatory networks in PBMCs to identify the key TFs of the immune response and identified specific motifs characteristic of goat T cells and monocytes in both control and case groups. For T cells, the case group showed motifs unique to BATF, KLF2, NFATC2, and CEBPB, whereas the control group showed motifs for YY1, PML, FOXP1, HIVEP2, ELK4, and SIN3A (Figure 5A). The transcription factor BATF, a highly conserved component of the AP-1 complex and a member of the basic leucine zipper ATF-like family, is expressed in a variety of immune cells, including T cells and dendritic cells. BATF plays a critical role in the development and functional maturation of these cells, underscoring its importance in orchestrating the immune system responses [28]. In monocytes, the case group goats displayed unique motifs for JUNB, NFIL3, JUND, ATF3, and CEBPB, while the control group goats had motifs for SREBF2, ELF2, PML, FLI1, and PRDM1 (Figure 5B). The CEBPB transcription factor, belonging to the C/EBP leucine zipper transcription factor family, is crucial for processes such as cell differentiation, immune responses, and inflammation [29]. In particular, it is essential for the production and survival of Ly6C monocytes; its absence results in decreased numbers and increased apoptosis of these cells [30]. By identifying specific transcription factors and their motifs, our study provides valuable insights into the molecular basis of the immune response in goats facing this parasitic challenge.

## 4. Discussion

In this study, we established a goat model artificially infected with *H. contortus* and validated the model establishment by a fecal egg count (FEC) test. However, there may be limitations because of the small sample size in the experiment. qPCR results showed significant upregulation of IL-5 and IL-6 genes in the case group, indicating that *H. contortus* infection induces a Th2 immune response in goats. IL-5 plays an important role in Th2 responses by activating eosinophils, which stimulates their survival, differentiation, and chemotaxis. IL-6, a multifunctional cytokine, is involved in the regulation of inflammatory responses [31]. Studies have shown that IL-6 expression is critical during the acute phase of helminth infections in mice, enhancing IgA secretion in the gastrointestinal mucosa and inducing fever during inflammation [32,33,34]. The response may play a crucial role in the immunological interaction between *H. contortus* and its host. Moreover, the temporal variation in immune gene expression may be associated with specific biological characteristics and infection stages of *H. contortus*.

Through scRNA-Seq analysis of goat PBMCs, we identified key cell subpopulations, including CD4 T cells, CD8 T cells, B cells, monocytes, and NK cells. Notably, the proportion of CD8 T cells and NK cells increased in the case group, while the monocyte fraction decreased, pointing to the distinct immunological roles these cells play during *H. contortus* infection. In our analysis of DEGs, we found a robust immune response in CD4 T cells, monocytes, and DCs. Specifically, genes like CXCR4 and NFKBIA were markedly upregulated in CD4 T cells of the case group. CXCR4 is known to facilitate the migration of mesenchymal stem cells (MSCs) to injured areas, accelerating wound healing, which may be associated with damage caused by *H. contortus* [35]. The upregulation of the inflammatory gene NFKBIA indicates that *H. contortus* infection induces an inflammatory response in the host. KEGG pathway enrichment analysis revealed significant enrichment in pathways such as the NOD-like receptor signaling pathway and Th1/Th2 cell differentiation, hinting that *H. contortus* infection may affect the Th1/Th2 differentiation of CD4 T cells and that pattern recognition receptors like NOD may also be involved in the immune response to parasites. Additionally, pathways associated with leishmaniasis, NF-κB signaling, and malaria were enriched in monocytes, underscoring the potential role of the NF-κB signaling pathway in the response to *H. contortus* infection. In DCs, the Toll-like receptor signaling pathway was significantly enriched, a pathway that has been extensively studied in helminth infections [36,37], although research related to *H. contortus* is comparatively limited and warrants further investigation.

To investigate the transcriptional regulatory mechanisms underlying the immune response of goats to *H. contortus*, we identified specific motifs unique to the case and control groups. The BATF motif was unique to the T cell subpopulations of goats in the *H. contortus*-infected group, and BATF plays a crucial role in the differentiation of Th17 cells [38]. Th17 cells, a subpopulation of CD4+ T cells, are involved in coordinating host defense and inflammatory responses in autoimmunity, with BATF binding to the promoters of genes related to Th17 differentiation, including IL-17, IL-21, and IL-22. Mice deficient in BATF show defects in Th17 differentiation [39]. Furthermore, BATF is also important in the activation and differentiation of CD8 T cells [40], and CD8 T cells lacking BATF show defective effector cell proliferation upon antigen encounter [41]. Previous research has indicated that the BATF transcription factor is key in the generation of Th2 cells and follicular helper T cells during helminth infections. Mice lacking BATF are unable to mount an effective Th2 response, and Th2 cells fail to produce necessary cytokines [42], suggesting that the BATF transcription factor may play a pivotal role in the host immune response to *H. contortus* infection.

Additionally, our ligand–receptor interaction analysis of scRNA-Seq data from case and control group goat PBMCs showed more interactions in the case group, indicating a more active cell communication state among PBMCs in goats infected with *H. contortus*. Notably, the analysis revealed significant differences in the interactions involving CD4 T and NK cells between the control and case groups. This observation suggests a potential role for these immune cells in the immune response against the parasite. We also found stronger interactions involving the MIF–(CD74 + CXCR4) ligand–receptor complex in the case group. MIF is a pleiotropic cytokine produced by various immune cells, including T cells, DCs, and B cells [41,43,44]. It can be directly secreted in response to various stimuli, such as infections and cytokine activation, and also acts as an “early response” cytokine that stimulates inflammatory reactions after secretion [45,46]. Reports indicate that mice deficient in MIF lack an effective innate immune response following infection with the *Nippostrongylus brasiliensis*. Moreover, there is an inability to undergo normal expansion of eosinophils and macrophages [47]. Marcon et al. discovered that the absence of MIF and pregnancy increase the severity of Toxoplasma gondii infection, leading to a decrease in intestinal villi and an increase in mucosal cell numbers, while also inducing a strong Th1 inflammatory response, which is further exacerbated by the lack of MIF [48]. Therefore, MIF may play a crucial role in the immune response of goats infected with *H. contortus*.

## 5. Conclusions

In this study, we first explored the immune response mechanisms of different subpopulations of goat PBMCs during *H. contortus* infection and identified critical genes, pathways, and transcription factors involved in the immune response. Additionally, we constructed a cellular communication network of goat PBMCs infected with *H. contortus*, providing valuable insights for a better understanding of the immune interaction mechanisms between goats and *H. contortus*.

## Figures and Tables

**Figure 1 cells-13-00842-f001:**
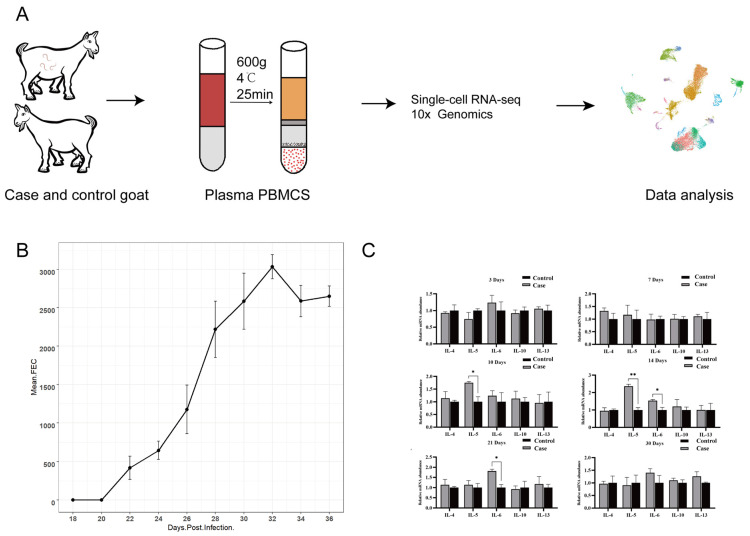
The change in EPG and immune gene expression. (**A**) Pipeline for case and control group processing and analysis; (**B**) EPG in goats infected with *H. contortus* at different times; (**C**) qPCR detection of immune gene expression in goat PBMCs at various time points postinfection with *H. contortus* (*, *p* < 0.05; **, *p* < 0.01).

**Figure 2 cells-13-00842-f002:**
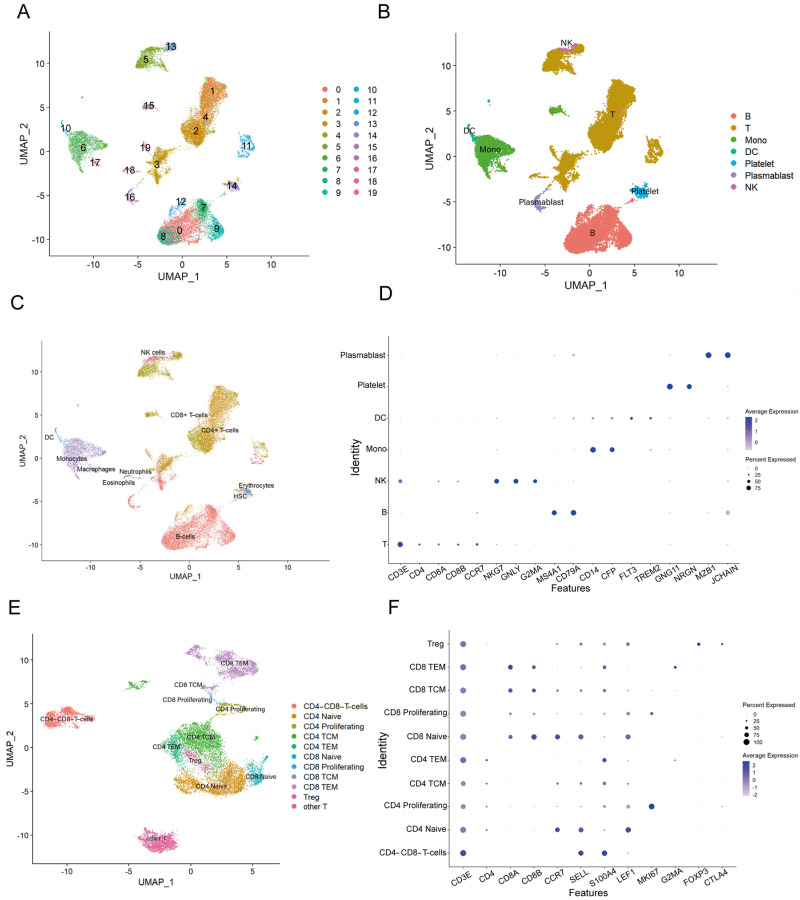
Annotation of cell subpopulations. (**A**) The clustered grouping plot of integrated samples; (**B**) UMAP plot annotated with SingleR; (**C**) UMAP plots illustrating the seven primary subpopulations of PBMCs; (**D**) bubble plots showing the expression and distribution of marker genes in each cell subpopulation of PBMCs; (**E**) UMAP plots depicting the 10 major subpopulations of T cells; (**F**) bubble plots depicting the expression and distribution of marker genes in each cell subpopulation of T cells.

**Figure 3 cells-13-00842-f003:**
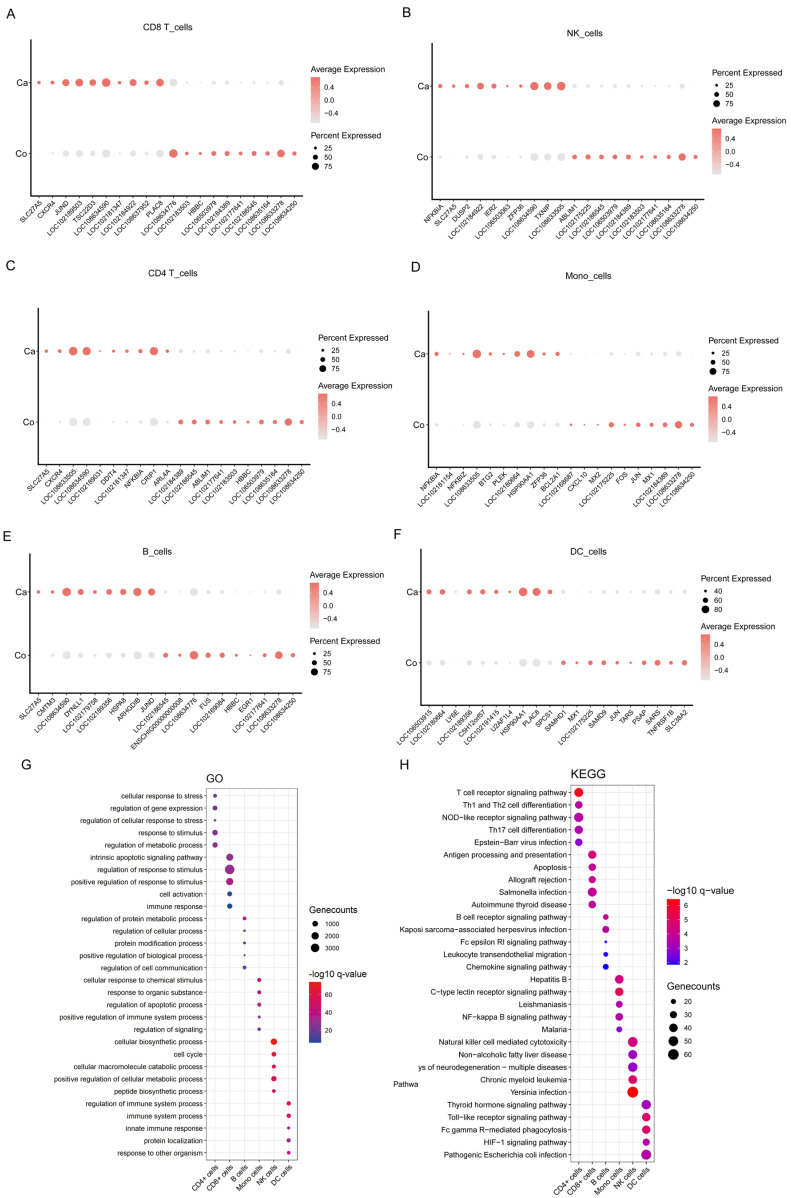
Differentially expressed genes in cell subpopulations of PBMCs from case and control groups. (**A**) The top 10 upregulated and downregulated genes in CD8 T cells; (**B**) the top 10 upregulated and downregulated genes in NK cells; (**C**) the top 10 upregulated and downregulated genes in CD4 T cells; (**D**) the top 10 upregulated and downregulated genes in monocytes; (**E**) the top 10 upregulated and downregulated genes in B cells; (**F**) the top 10 upregulated and downregulated genes in DCs; (**G**) GO enrichment analysis of DEGs; (**H**) KEGG enrichment analysis of DEGs.

**Figure 4 cells-13-00842-f004:**
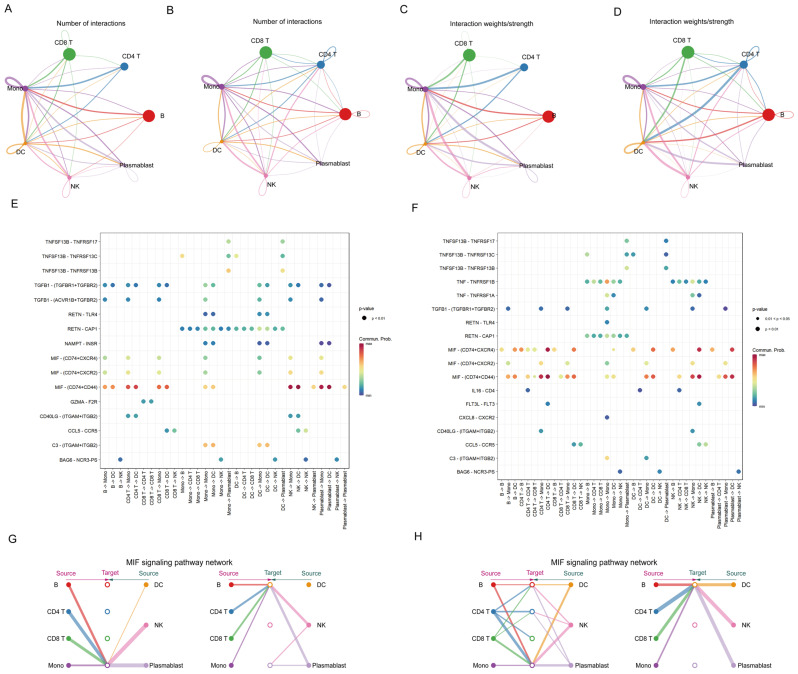
Cell communication among subpopulations of goat PBMCs. (**A**) Number of interactions among subpopulations of PBMCs in the control group of goats; (**B**) number of interactions among subpopulations of PBMCs in the case group of goats; (**C**) the strength of interaction among subpopulations of goat PBMCs in the control group calculated by summing the probability values; (**D**) the strength of interactions among subpopulations in the case group of goat PBMCs; (**E**) cell communication among subpopulations of goat PBMCs in the control group: the *x*-axis represents cell combinations, and the *y*-axis represents ligand–receptor pairs that participate in cell–cell interactions in the communication network; (**F**) goat PBMC subpopulation cell communication in the case group; (**G**) hierarchical plot showing inferred cell interactions in the control group: solid and open circles refer to the source and target, respectively, while lines represent intercellular interactions, their thickness is proportional to the communication probability of the cell interaction, and; line colors correspond to the signaling source; (**H**) hierarchical plot showing inferred cell interactions in the case group.

**Figure 5 cells-13-00842-f005:**
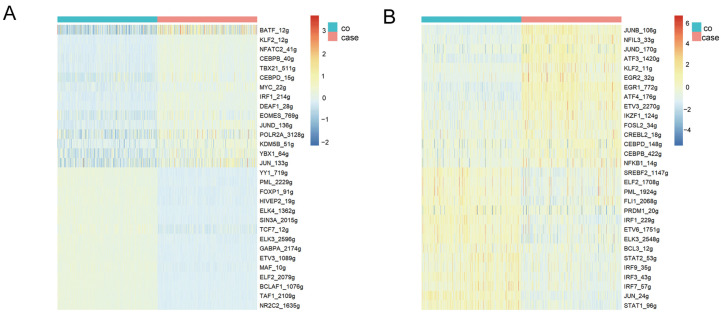
Comparative analysis of transcription factor regulation. (**A**) The top 15 transcription factors with the greatest differential upregulation or downregulation between the control group and the case group of T cells; (**B**) the top 15 transcription factors with the greatest differential upregulation and downregulation between the control group and the case group of monocytes.

## Data Availability

The sequencing data are deposited in the Genome Sequence Archive in BIG Data Center, Beijing Institute of Genomics (BIG, http://gsa.big.ac.cn), Chinese Academy of Sciences (project accession no. PRJCA024514 and GSA accession no. CRA015490; https://ngdc.cncb.ac.cn/gsa/s/8nPa9TGS). We included other relevant data in this original manuscript file and/or in the Appendix A. Nevertheless, the corresponding author will provide additional data related to these findings upon reasonable request.

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
