# Peer review of "Single-Cell Transcriptomic Profiling Unveils Dynamic Immune Cell Responses during Haemonchus contortus Infection"

_cells, 2024, doi:10.3390/cells13100842_

Round 1

Reviewer 1 Report

Comments and Suggestions for Authors

Comments on the article ID: cells-2983094 entitled “Single-Cell transcriptomic profiling unveils dynamic immune cell responses during Haemonchus contortus infection”, submitted by Wang et colleagues to CELLS.

Considering that Haemonchus contortus is one of the most pathogenic parasites of ruminants, particularly grazing goats and sheep, this study appears original and useful in that it provides important information to supplement the knowledge on this gastrointestinal nematode of ruminants. The article is suitable for the journal and overall is not poorly written. However, the authors should consider the comments below on different sections of the article to improve it.

-  Some parts of the manuscript, especially the Results section, should be simplified. In particular, Subsections 2.1, 2.2, 2.3 and 2.4 need to be revised because in all four subsections the introductory part and other parts of the text deal with topics that should be described in the Materials and Methods section. Simplify these subsections and check the entire text by removing all parts relating to section 5. Materials and Methods. 

- Line 37: The text incorrectly states that Haemonchus contortus “primarily parasitises the rumen of various ruminants”. Haemonchus contortus is a gastrointestinal nematode widely distributed in tropical, subtropical, and warm temperate regions that localizes in the “abomasum” of several ruminant species. Please correct.

-    Line 42: To reinforce the sentence, please cite this 2024 study: Castagna F, Bava R, Palma E, Morittu V, Spina A, Ceniti C, Lupia C, Cringoli G, Rinaldi L, Bosco A, Ruga S, Britti D and Musella V (2024) Effect of pomegranate (Punica granatum) anthelmintic treatment on milk production in dairy sheep naturally infected with gastrointestinal nematodes. Front. Vet. Sci. 11:1347151.doi:10.3389/fvets 2024.1347151.

-  Line 60: The acronym PRR has already been given in parentheses in line 53. In subsequent parts of the text use only the acronym. Check the entire document and correct the other acronyms as well.

-  Line 65: The year of scientific publication should be given after Lu et al. Check the paper if there is the same problem for other studies and correct it.

-  Line 70: In this section, the acronym PBMCs should be given in parentheses the first time “peripheral blood mononuclear cells” are written.

-  Line 91: The correct wording is "eggs per gram (EPG) of feces." Please correct and state only the acronym EPG later. Check for similar inconsistencies in the text and tables and correct them.

-  Lines 180-183: Why are you talking about hookworm? Please clarify this sentence.

- Line 227: H. contortus should be written in italics, please correct. Check and correct the binomial nomenclature also in the bibliography. 

Conclusion: “The manuscript may be accepted after major revisions before resubmission”.

Comments on the Quality of English Language

Overall it's well written, review the punctuation.

Reviewer 2 Report

Comments and Suggestions for Authors

This manuscript is one of the first to report on a single-cell transcriptomic analysis of goat PBMC in animals infected with H. contortus as compared to controlled. It is well-written and will serve as a starting point for others to target genes involved in the response to this parasite in order to select animals more resilient to infection. The greatest weakness is the limited number of animals used in the experiment.  The figures included are much to small to read, therefor it is impossible to tell if the written results concur with the figures.

L22:  "identified" should be "identifying"

L40-43: these sentences are redundant and should be combined.

L102-121: should be part of the discussion rather than the results section.

L183: this is not a manuscript about hookworms - is this an error?

Section 5.1: please describe the animals used in more detail. How old were they? How were they kept parasite-free? Were they from the same herd? Were they related? Were they selected from a larger pool of animals for any specific reason?

Round 2

Reviewer 1 Report

Comments and Suggestions for Authors

The authors greatly improved the manuscript and made the requested change. It is requested to add on line 104, after (EPG), the word "of feces".

Author Response

Thank you very much for your review and valuable comments on our manuscript. We very much appreciate your recognition of the study and fully agree with the points you have raised.

In line 104, we have made correction according to the reviewer’s comments . The original sentence is "eggs per gram (EPG) were determined bi-daily using the  McMaster egg counting technique".

The revised sentence is “eggs per gram (EPG) of feces were determined bi-daily using the  McMaster egg counting technique".